# Characterisation of *Plasmodium vivax* lactate dehydrogenase dynamics in *P. vivax* infections
Pengxing Cao[1,9], Steven Kho[2,3,9], Matthew J. Grigg [2], Bridget E. Barber [4], Kim A. Piera[2], Timothy William[2], Jeanne R. Poespoprodjo[3,5], Ihn Kyung Jang[6], Julie A. Simpson[7], James M. McCaw[1,7], Nicholas M. Anstey[2,10], James S. McCarthy [4,8,10] ✉ & Sumudu Britton [4,10] ✉

*Plasmodium vivax* lactate dehydrogenase (PvLDH) is an essential enzyme in the glycolytic pathway of *P. vivax*. It is widely used as a diagnostic biomarker and a measure of total-body parasite biomass in vivax malaria. However, the dynamics of PvLDH remains poorly understood. Here, we developed mathematical models that capture parasite and matrix PvLDH dynamics in ex vivo culture and the human host. We estimated key biological parameters characterising in vivo PvLDH dynamics based on longitudinal data of parasitemia and PvLDH concentration collected from *P. vivax*-infected humans, with the estimates informed by the ex vivo data as prior knowledge in a Bayesian hierarchical framework. We found that the in vivo accumulation rate of intraerythrocytic PvLDH peaks at 10–20 h post-invasion (late ring stage) with a median estimate of intraerythrocytic PvLDH mass at the end of the life cycle to be $9.4 \times 10^{-3}$ng. We also found that the median estimate of in vivo PvLDH half-life was approximately 21.9 h. Our findings provide a foundation with which to advance our quantitative understanding of *P. vivax* biology and will facilitate the improvement of PvLDH-based diagnostic tools.

Malaria is a vector-borne disease, with 247 million malaria cases and 619,000 deaths estimated worldwide in 2021[1]. *Plasmodium vivax* is the most geographically widespread species causing malaria, putting billions of people at risk of infection[2,3]. Although diagnosis of malaria by light microscopy remains the primary method, rapid diagnostic tests (RDTs) that detect *Plasmodium*-specific biomarkers present in the peripheral blood are widely used. *Plasmodium* lactate dehydrogenase (pLDH) is an essential enzyme in the glycolytic pathway of malaria parasites, and was one of the first parasite biomarkers identified in the blood of *Plasmodium*-infected patients[4,5]. The development of RDTs that use *P. vivax*-specific monoclonal antibodies to target pLDH (PvLDH) has enabled species-specific detection of vivax infection without microscopy in areas co-endemic for *P. falciparum* and *P. vivax*[6]. Quantification of plasma PvLDH has also been used as a measure of *P. vivax* biomass in clinical studies of uncomplicated and severe vivax malaria[7,8].

An understanding of the quantitative dynamics of a given biomarker facilitates its application as a diagnostic tool[9] and may reveal novel applications. Compared with the *P. falciparum*-specific biomarker, histidine-rich protein-2 (HRP2), the dynamics of which have been extensively studied and modelled[9–12], the dynamics of PvLDH remains poorly characterised. Furthermore, recent human studies have confirmed the accumulation of *P. vivax* parasites in reticulocyte-rich tissues outside the circulation, such as the spleen[13,14], and to a lesser extent the bone marrow[15,16], with the splenic reservoir accounting for more than 98% of total-body *P. vivax* biomass in chronic infections[13]. Nevertheless, in the absence of invasive tissue sampling, quantification of any parasite biomass that is outside peripheral circulation remains an ongoing challenge.

This highlights the need for better understanding the dynamics of *P. vivax* specific biomarkers. However, no mathematical methods for quantifying the biological parameters governing PvLDH dynamics based on in

[1]School of Mathematics and Statistics, University of Melbourne, Melbourne, VIC, Australia. [2]Global and Tropical Health Division, Menzies School of Health Research and Charles Darwin University, Darwin, NT, Australia. [3]Papuan Community Health and Development Foundation, Timika, Papua, Indonesia. [4]QIMR Berghofer Medical Research Institute, Brisbane, QLD, Australia. [5]Department of Pediatrics, Timika General Hospital, Timika, Papua, Indonesia. [6]Diagnostics Program, PATH, Seattle, WA, USA. [7]Centre for Epidemiology and Biostatistics, Melbourne School of Population and Global Health, University of Melbourne, Melbourne, VIC, Australia. [8]Department of Infectious Diseases, Melbourne Medical School, Melbourne, VIC, Australia. [9]These authors contributed equally: Pengxing Cao, Steven Kho. [10]These authors jointly supervised this work: Nicholas M. Anstey, James S. McCarthy, Sumudu Britton. ✉e-mail: james.mccarthy@unimelb.edu.au; sumudu.britton@health.qld.gov.au

vivo data are available. PvLDH data collected from vivax malaria patients and participants in human challenge studies where parasite kinetics are measured in a controlled setting provide an opportunity for the characterisation of both ex vivo and in vivo PvLDH dynamics using mathematical modelling, and allow us to infer estimates for biological parameters that are difficult to measure by direct sampling.

In this study, we quantified the dynamics of PvLDH by fitting two mathematical models to experimental data. The first model (referred to as the ex vivo model) captures the ex vivo dynamics of parasites and PvLDH and was fitted to a set of longitudinal measurements of parasite count and PvLDH concentration in two short-term ex vivo cultures of *P. vivax* patient isolates. The parameters identified from the ex vivo model then served as prior knowledge for the second model (referred to as the within-host model), which captured in vivo parasite and PvLDH dynamics that was fitted to longitudinal data of parasitemia and whole blood PvLDH concentration from eight adults experimentally-infected with *P. vivax* in a volunteer infection study (VIS). Both model fittings were conducted using Bayesian hierarchical inference, which is a statistical method for generating the posterior distribution of model parameters that quantify the mean properties of the examined cohort, based on both experimental data and any prior knowledge of parameter distribution gained from other independent experimental studies. We report the ex vivo and in vivo estimates of two fundamental parameters that characterise PvLDH dynamics. Firstly, intraerythrocytic PvLDH, which describes the average net accumulation of PvLDH mass inside a parasitized red blood cell (pRBC) over an asexual life

cycle and is a function of parasite age. Secondly, the PvLDH decay rate or half-life, which quantifies how quickly PvLDH decays after release from ruptured pRBCs.

## Results

### Quantification of ex vivo PvLDH dynamics

The ex vivo model was fitted to both parasite count data derived by microscopy and ELISA-based PvLDH measurements collected in two 54-hour ex vivo cultures of *P. vivax* from two Indonesian patients using Bayesian hierarchical inference (see Methods for further details about the model and the ex vivo experiments). PvLDH levels in culture media measured at six-hour intervals indicate the amount of PvLDH released by ruptured pRBCs, while PvLDH levels in red blood cell (RBC) pellets indicate the intracellular PvLDH accumulated in the pRBCs. Figure 1a–d present the results of the model fitting, and show that the ex vivo model captures the trends of the measured data, as visualised by the median and 95% prediction interval (PI) of parasite count and PvLDH levels over time. We observed a rapid increase in PvLDH in culture media immediately after 0 h in patient 2 (as opposed to the gradual increase in media PvLDH for Patient 1); this could be explained by the presence and rupturing of a higher fraction of schizonts at the start of the culture in patient 2 compared to patient 1 (2% vs. 0% by microscopy; 5.5% vs. 0.1% by the median model prediction). Data points in Fig. 1e, f present the proportion of parasites at each stage of the asexual cycle in the culture as determined by microscopy. At the start of culture, microscopy indicated that parasites were 91% and 93% ring stages

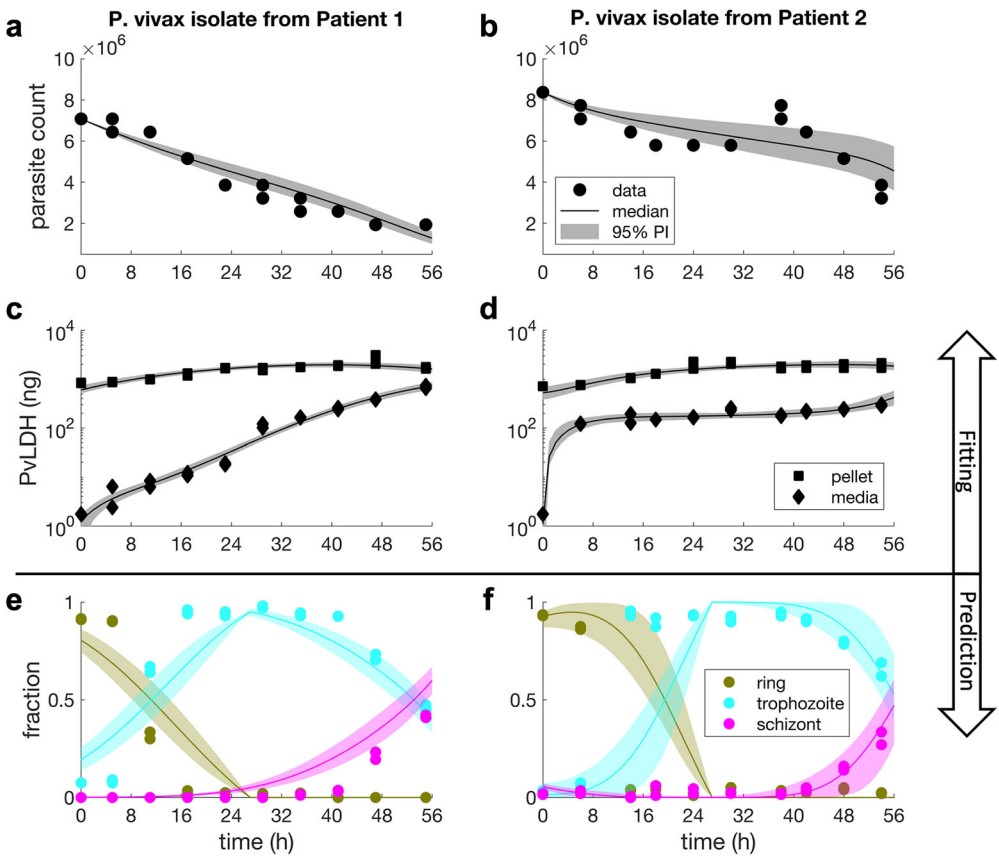

**Fig. 1 | Results of ex vivo model fitting and model predictions.** Panel **a–d** show the results of fitting the ex vivo model to the culture data of P. vivax isolates from two malaria patients in Timika, Indonesia. The solid dots are experimental measurements of parasite counts in the cultures and the solid squares and diamonds are experimental measurements of PvLDH in the culture pellet and media, respectively (note that PvLDH concentrations were measured and then converted to mass based on the volume of the culture; see Methods for details). Model fits are shown by the median and 95% prediction interval (PI) of the model-predicted distributions of

parasite count and PvLDH levels (in nanogram) at different time points (in hours). The calculation of the median and the 95% PI is described in the Methods. Panel **e** and **f** show the model-predicted fractions of parasites at different stages of the lifecycle, which are shown by the median predictions (colour curves) and associated 95% PIs. Experimental measurements of the fractions of rings, trophozoites and schizonts are also shown in the figures for comparison (solid-coloured dots). Experimental measurements were conducted in duplicate at each timepoint.

(in patient 1 and 2, respectively); these matured into >90% trophozoites by 14—17 hours, and finally into schizonts (41% and 30% of parasites in patient 1 and 2, respectively) by the end of the culture. The fitted model was able to predict the fraction of parasites at various asexual developmental stages, and the predicted time series (visualised by the 95% PIs with median prediction curves shown in Fig. 1e, f) were relatively consistent with experimental measurements, further supporting the ability of the ex vivo model to quantitatively capture the dynamics of the ex vivo system.

Using the posterior samples of the population mean parameters obtained from the model fitting (marginal distributions of the samples are shown in Supplementary Fig. S1 in the Supplementary Information), we estimated the ex vivo age-dependent accumulation of intraerythrocytic PvLDH and ex vivo PvLDH decay rate. Data modelling using a Hill function (Eq. 6 in the Methods) indicated that the median level of intraerythrocytic PvLDH was 2 to 3 orders of magnitude higher in mature stages (late trophozoite and schizont) compared to early stages (early and intermediate rings) (Fig. 2a). This is consistent with earlier findings that LDH levels are higher in blood samples where mature parasites are present[4]. The maximum PvLDH per parasite at the end of the asexual life cycle was $5.1 \times 10^{-4}$ ng (95% PI: $1.8 \times 10^{-4}$—$2.7 \times 10^{-3}$) (Fig. 2a). The observed increase in PvLDH level as parasites age, suggests an accumulation of PvLDH produced by the parasite within the infected erythrocyte, and is in keeping with the slow increase in pellet PvLDH observed in the first 20–30 hours of the cultures (Fig. 1c, d).

The ex vivo PvLDH decay rate indicates the speed of decay of PvLDH in the culture media. Based on the marginal posterior distribution of the population mean of the ex vivo PvLDH decay rate (shown in Supplementary Fig. S1 in the Supplementary Information), the ex vivo PvLDH decay rate was estimated to be $0.0106\,\text{h}^{-1}$ (equivalent to a half-life of 65.3 h) and a 95% credible interval (CrI) of $0.0098$—$0.0114\,\text{h}^{-1}$ (60.8—70.7 h). We also present the net PvLDH accumulation rate, calculated from the derivative of the intraerythrocytic PvLDH concentration, an indicator of the rate of accumulation of PvLDH inside the infected red cell (as a net result of production partially offset by degradation) (Fig. 2b). The ex vivo model predicts that the net rate of intraerythrocytic PvLDH accumulation peaks at approximately 20 h from the start of an asexual life cycle during which parasites are at the ring stage (Fig. 2b).

## Quantification of in vivo PvLDH dynamics

Based on a similar framework used to construct the ex vivo model, we developed a within-host model to capture the in vivo dynamics of parasite growth and PvLDH turnover, and fitted the model to the parasitemia and whole blood PvLDH concentration data from eight human volunteers experimentally-infected with *P. vivax* isolates from India in a VIS (details about the VIS, the model and fitting method are provided in the Methods). To refine the within-host model, priors of two model parameters that

determine the intraerythrocytic PvLDH level (i.e., $c_1$ and $c_2$ in the within-host model; see Table 2 in the Methods) were proposed based on the posterior distributions of the ex vivo model (see the green dotted curves in Supplementary Fig. S1 in the Supplementary Information). Fig. 3a–d show the model fit and experimental data for two representative volunteers (R009 and R010; results for all eight volunteers are shown in Supplementary Fig. S2 in the Supplementary Information). The model reproduced the synchronous stepwise growth pattern of *P. vivax* parasitemia and PvLDH production during the pre-treatment phase, as well as the decline in parasitemia and PvLDH levels after chloroquine treatment on day 10 (Fig. 3a–d). We also used the model to predict the fraction of rings, trophozoites and schizonts over the period of observation, similar to the ex vivo study (albeit with no data for validation) (Fig. 3e, f). Consistent with the well-established life-cycle dynamics of *Plasmodium* infection, we found that the modelled distribution of parasite numbers across the life-cycle stages oscillates, and that the predicted fraction of rings rapidly increases at the same time as a stepwise increase in parasitemia during the pre-treatment phase, timing that is consistent with schizont rupture.

We next calculated the in vivo intraerythrocytic PvLDH level using data from posterior distributions (for further details see Methods and Supplementary Fig. S3 in the Supplementary Information for the marginal posterior distributions of population mean parameters). The estimated in vivo intraerythrocytic PvLDH level reached a median value of $9.2 \times 10^{-3}$ (95% PI: $2.1 \times 10^{-3}$—$1.4 \times 10^{-2}$) ng PvLDH per parasite at the end of the lifecycle (Fig. 4a). In addition, the in vivo net intraerythrocytic PvLDH accumulation rate (Fig. 4b) peaked at 10–20 h post-invasion (late ring stage).

Based on the marginal posterior distribution of the population mean of the in vivo decay rate ($\lambda$ in Supplementary Fig. S3 in the Supplementary Information), we estimate the in vivo PvLDH decay rate in the within-host model to have a median of $0.0316\,\text{h}^{-1}$ (equivalent to a half-life of 21.9 h) and a 95% CrI of $0.0232$–$0.0416\,\text{h}^{-1}$ (16.7–29.9 h).

## Discussion

In this study, we modelled ex vivo and in vivo experimental data of parasitemia and PvLDH and estimated key biological parameters that characterise the dynamics of PvLDH using Bayesian hierarchical inference. We derived two important parameters, the intraerythrocytic PvLDH level per pRBC (a function of parasite age) and the PvLDH decay rate. We found that in vivo and ex vivo estimates differed significantly, with intraerythrocytic PvLDH mass estimates in vivo at least ten-fold higher across the whole asexual life cycle than those estimated in the ex vivo system. At the end of the life cycle, the median intraerythrocytic PvLDH mass estimated by the in vivo and ex vivo models was $9.4 \times 10^{-3}$ and $5.1 \times 10^{-4}$ ng per pRBC, respectively. The rate of intraerythrocytic PvLDH accumulation was fastest during the ring stage in both systems. Similarly, ex vivo and in vivo estimates of PvLDH

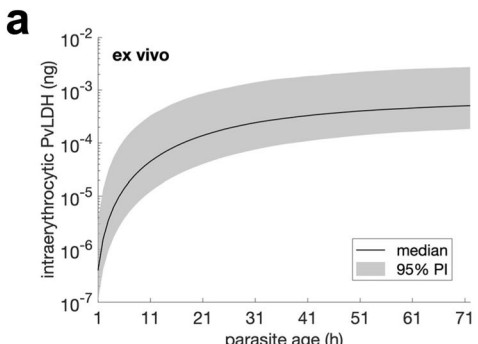
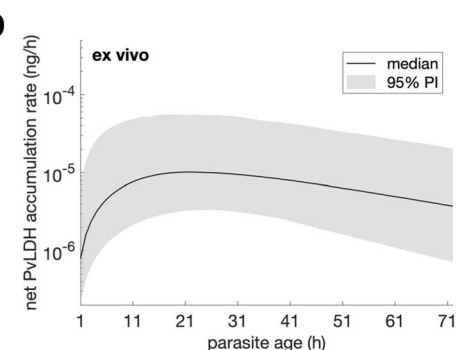

**Fig. 2 | Estimates of ex vivo intraerythrocytic PvLDH.** Estimated level of ex vivo intraerythrocytic PvLDH (ng) and net intraerythrocytic PvLDH accumulation rate (ng/h) over an ex vivo asexual life cycle. Note that the length of the ex vivo life cycle was extended to 72 h due to experimental conditions (see Methods for further details). The net intraerythrocytic PvLDH accumulation rate shown in **b** is the rate-of-change (the derivative) of the ex vivo intraerythrocytic PvLDH shown in **a**. Details of the calculation of the two quantities and their median and 95% prediction interval (PI) based on the posterior samples of population mean parameters are provided in the Methods.

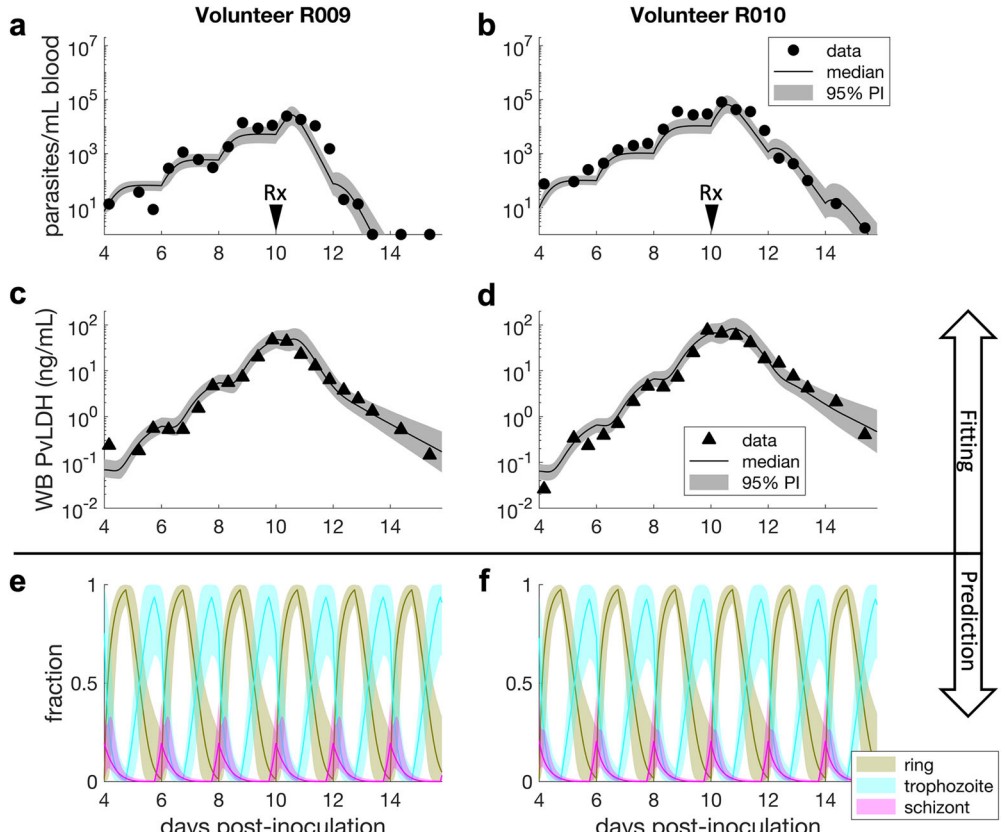

**Fig. 3 | Results of within-host model fitting and predictions in VIS.** Panels **a**–**d** show representative plots from two individuals after fitting the within-host model to parasitemia and whole blood (WB) PvLDH experimental data obtained from eight human volunteers experimentally-infected with P. vivax in a VIS. Model fits are shown by the median and 95% prediction interval (PI) of model-predicted distributions of parasitemia and WB PvLDH concentration at different time points. The calculation of the median and 95% PI is described in the Methods. The two volunteers were treated with a standard course of oral chloroquine on day 10 post-inoculation (indicated by the arrows labelled with Rx). Panel **e** and **f** show the model-predicted fractions of asexual developmental stages, illustrated by the median predictions (colour curves) and associated 95% PIs. For volunteer R009, some post-treatment parasitemia data are truncated at the lower detection limit of 1 parasite/mL.

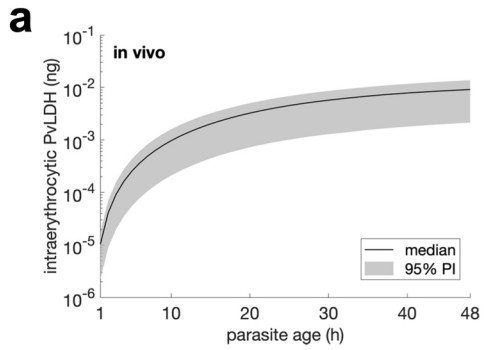
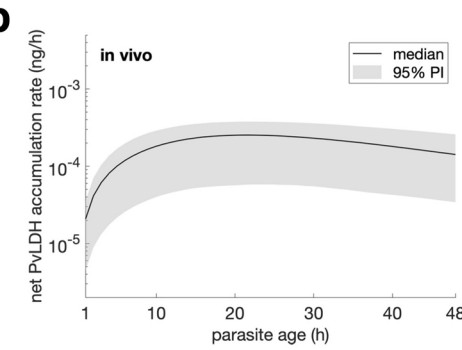

**Fig. 4 | Estimates of in vivo intraerythrocytic PvLDH.** Estimated in vivo intraerythrocytic PvLDH level and the net accumulation rate of intraerythrocytic PvLDH over an in vivo asexual life cycle. The net intraerythrocytic PvLDH accumulation rate shown in **b** is the rate-of-change (the derivative) of the intraerythrocytic PvLDH shown in **a**. The curve and shaded area represent the median and 95% prediction interval (PI) of model predictions.

decay rates were distinct, with PvLDH clearance from the blood three times faster than the decay rate in culture media. To our knowledge, this is the first study characterising both ex vivo and in vivo PvLDH dynamics under a single modelling framework and provides an important foundation to advance our understanding of *P. vivax* biology and the development of PvLDH-based diagnostic tools.

The finding that in vivo estimates of intraerythrocytic PvLDH mass were at least ten-fold higher than ex vivo estimates was unexpected. Growing *P. vivax* in culture remains a major challenge[17] as reflected by the presence of morphologically unhealthy parasites and prolonged ex vivo maturation that is commonly seen in such experiments. In comparison to an open in vivo system where nutrient and serum levels are high with constant turnover of toxins, we hypothesise that metabolic activity of parasites in our closed culture system may be impaired, including PvLDH production, due to factors such as reduced serum levels and build-up of toxic by-products. Other biological conditions or factors that are present/absent in vivo versus ex vivo may further modulate the production of intraerythrocytic PvLDH. Similarly, the PvLDH decay rate, which measures the rate of decline in

PvLDH in either the peripheral blood (in vivo) or the culture media (ex vivo), was approximately three times faster in vivo than ex vivo. We hypothesise that specific processes/factors that are present in vivo but absent in cell culture, such as protein immune-complex formation, metabolic degradation and renal clearance, may lead to faster in vivo clearance of PvLDH from peripheral blood.

Our estimates of intraerythrocytic PvLDH level as a function of parasite age provide a comprehensive estimate of PvLDH accumulation inside a pRBC across multiple asexual developmental stages. Our ex vivo estimates of the intraerythrocytic PvLDH mass and accumulation rate were qualitatively consistent with earlier findings by Barber et al. who identified a rapid increase in PvLDH production during the first 6–12 hours of the life cycle followed by a slow increase or saturation for the remainder of life cycle[7]. A quantitative comparison of their results with our data was not possible due to the lack of a demonstrably synchronised parasite culture in the earlier study. Our estimates of the pattern of intraerythrocytic PvLDH accumulation over time are also consistent with transcriptomic data whereby levels of *P. vivax* LDH mRNA peak across the 6 h to 24 h interval, followed by a gradual decline until the end of the lifecycle[18].

This model provides a platform to further evaluate the utility of *P. vivax* biomarkers for non-invasively quantifying the hidden parasite biomass seen with this infection[7,8,13,14]. This would require the availability of sufficiently sensitive and accurate assays for measuring PvLDH, and other potential *P. vivax* specific biomarkers, in plasma as well as whole blood. Serial measurement of plasma and whole blood PvLDH and parasitemia in patients with symptomatic malaria infection as well as in patients with asymptomatic sub-patent infection would provide additional key data to improve the accuracy of the model.

Some of the potential clinical applications of modelling frameworks such as this is that calculation of key PvLDH dynamics is also a key step in improving the development and interpretation of PvLDH based RDTs. Parameters such as clearance of PvLDH can advise minimal limits of detection for target product profiles for the development of new RDTs in order to maintain optimal sensitivity. Better knowledge of biomarker dynamics can also guide interpretation of RDT results. For example, whether a positive RDT after a recently treated infection is likely to be new or recrudescent infection rather than being consistent with ongoing clearance of the previous infection.

Our study has several limitations. Our mathematical model does not account for the presence of gametocytes. This is unlikely to have a material impact on the inferred parameters governing the asexual replication cycle as the sexual commitment rate per cycle is less than 1%[19]. Secondly, we assumed that the intraerythrocytic PvLDH could only be released to plasma upon rupture of viable or dead pRBC. This assumed mechanism of PvLDH release was sufficient to reproduce the ex vivo experimental observation, but we cannot exclude gradual release and there have been studies showing that *P. falciparum*-specific LDH can be released from intact pRBC by extracellular vesicles[20]. This has not been reported for *P. vivax*. Thirdly, we were unable to test whether parasite genetic factors also play a role in varying PvLDH production rates (i.e., Indian VIS isolate vs. Indonesian clinical isolates). However, recent work suggests PvLDH is genetically well-conserved in global populations[21], and that the pattern of our modelling estimates was consistent with transcriptomic data from Thai *P. vivax* isolates[18], suggesting minimal genetic variation. Fourthly, we found that the ex vivo model predictions did not precisely reproduce some of the parasite staging behaviour seen in the data, such as the ring-to-trophozoite transition in both patients. This suggests that the staging process were either not captured by the current model (which was not designed for studying the staging process) or not informed by the parasitemia data. Nevertheless, this did not affect the quality of the primary data on model-predicted PvLDH dynamics in our study. Further work is warranted to investigate the staging process which may require the development of new models and additional experimental validation. Lastly, the estimates of PvLDH dynamics generated based on our experimental set up may vary between *P. vivax* geographical locations and experimental conditions. Promising advancements

to enable continuous *P. vivax* culture systems are being made and will enable standardisation of future ex vivo studies of this *Plasmodium* species.

In conclusion, we have established a modelling framework for characterising PvLDH dynamics. We provide both ex vivo and in vivo estimates of intraerythrocytic PvLDH levels across the *P. vivax* asexual life cycle and provide estimates of its decay rate. Our work advances our quantitative understanding of PvLDH dynamics and pave the way for the better understanding of *P. vivax* pathobiology and the development of novel methods/tools based on the biomarker to support diagnosis and control of *P. vivax* infections.

## Methods
### Study design
The mathematical model of *P. vivax* and PvLDH dynamics was developed in a two-step process. First, we developed the ex vivo model which was fitted to the ex vivo experimental data derived from *P. vivax* culture and serial measurements of PvLDH at different lifecycle stages during the ex vivo culture. Parameters identified by this model were then combined with longitudinal parasitemia and whole blood PvLDH measurements from a *P. vivax* malaria volunteer infection study (VIS) to develop the within-host (in vivo) model in order to calculate key biomarker dynamics.

### *Plasmodium vivax* culture and microscopy
The ex vivo intraerythrocytic PvLDH level was estimated based on experimental data from ex vivo *P. vivax* cultures. Parasite isolates were obtained from malaria patients with *P. vivax* mono-infection as part of a larger surveillance study in Timika, Papua, Indonesia. Written informed consent was obtained for collection of peripheral venous blood. The work was approved by the Human Research Ethics Committees of Gadjah Mada University, Indonesia (KE/FK/0505/EC/2019), and Menzies School of Health Research, Australia (HREC 10-1937). Data from two experiments with optimal cultures, starting parasitemia (>1%) and initial staging (>90% young rings) were selected for the modelling work to satisfy the compatibility between the data and our mathematical model and the inclusion of all developmental stages over the asexual lifecycle.

Fresh Indonesian *P. vivax* field isolates were cultured at 10% haematocrit in warmed McCoy's 5 A media supplemented with HEPES, gentamycin, D-glucose, L-glutamine and human serum, as previously described[4,22]. For each experiment, 500 µL of culture suspension was added into each well of a 48-well plate and incubated in a candle jar at 37 °C within 2 hours of blood collection. The entire volume of two wells were sampled every 6 hours, resulting in a total of 10 timepoints being collected over a 54-hour culture period. Exact timepoints were recorded to the closest half hour. At each timepoint, suspensions from the two duplicate wells were centrifuged at low speed to gently pellet the red cells. Culture media and red cell pellets were separated and immediately frozen at −80 °C. Thick and thin blood smears were prepared for each well at each timepoint and stained with 3% Giemsa solution for 45 minutes at room temperature. Smears were examined by a WHO-certified expert microscopist. Parasite levels were calculated in the thin smear from the number of parasites seen in at least 1000 red cells and converted to total parasite count per well using the average number of red cells per well $6.44 \times 10^8$ estimated based on automated red cell counts and haematocrit data from 19 patients with uncomplicated vivax malaria in a previous cohort[23]. Healthy parasites were morphologically staged into rings, trophozoites, schizonts and gametocytes, and the fractions of ring, trophozoite and schizont stages are shown in Fig. 1 for the purpose of determining the length of of in vitro life cycle.

### Inhouse PvLDH ELISA
The concentration of PvLDH was measured in frozen culture media and RBC pellets using an inhouse ELISA as previously described[7]. RBC pellets were freeze-thawed four times prior to the assay. ELISA plates were read on a GloMax plate reader (Promega, Wisconsin, US). Data were converted from concentration to total mass based on the volume of media/pellet.

PvLDH in the media was a measure of PvLDH released by the parasites, while PvLDH in the pellets was a measure of intracellular PvLDH produced by the parasites.

### In vitro PvLDH decay

The in vitro PvLDH decay rate was estimated by fitting an exponential decay model (Eq. 1 below) to the in vitro measurements of human PvLDH decay over time. Human PvLDH from a plasma sample of known PvLDH concentration was introduced to a 10%-hematocrit suspension of healthy uninfected RBCs and cultured in a 48-well plate without parasites under the same conditions described above. Culture media was collected from duplicate wells and frozen at time zero, 30 minutes, then every 6–8 hours over a total of 54 hours. PvLDH concentration in the media at each timepoint was measured by inhouse ELISA as described above and presented in nanograms per mL (Supplementary Fig. S4 in the Supplementary Information). The exponential decay model is given by

$$L_{vitro}(t) = L_{vitro}(0)e^{-\lambda'' t}, \tag{1}$$

where $L_{vitro}(t)$ represents the in vitro PvLDH level at time $t$ since the first measurement and $\lambda''$ is the in vitro PvLDH decay rate (note that we use the double prime superscript to differentiate it from the ex vivo and in vivo decay rates introduced later).

Model fitting was conducted in Bayesian framework and samples of the target posterior distribution were generated using Hamiltonian Monte Carlo. The prior distributions for $L_{vitro}(0)$ and $\lambda''$ were uniform distributions U(500, 1000) and U(0, 0.1), respectively. Four Markov chains with starting points randomly chosen from the prior distributions were generated and each chain returned (after a burn-in period of 1000 samples) 1000 samples which were used to estimate model parameters and produce marginal posterior distributions. Model fitting was implemented in R (version 4.0.5) and Stan (version 2.21.5). Data and computer code are publicly available at https://doi.org/10.26188/23256413.v1.

Posterior predictive checks of the model fitting and the marginal posterior distributions of $L_{vitro}(0)$ (initial PvLDH) and $\lambda''$ are shown in Supplementary Figs. S4 and S5, respectively. The posterior samples of $\lambda''$ can be fitted by a normal distribution with a mean of 0.0106 and a standard deviation of $4.1565 \times 10^{-4}$. This normal distribution is used as the prior distribution for the ex vivo decay rate of PvLDH in the ex vivo model which is introduced in the next section.

### Ex vivo mathematical model of *P. vivax* and PvLDH dynamics

We developed a mathematical model of *P. vivax* and PvLDH dynamics in the ex vivo culture system (referred to as the ex vivo model) which was fitted to the ex vivo experimental data. The parasite maturation dynamics in the ex vivo model is structurally similar to the within-host model (described below) except that there is no parasite replication due to the lack of reticulocytes in the culture:

$$P'(a, t) = P'(a - 1, t - 1)e^{-\delta'}, \quad a = 2, 3, \ldots, a'_L. \tag{2}$$

We use $P'(a, t)$ (with a prime) to represent the parasite number in the culture and $P'(1, t) = 0$ for all $t > 0$. Since we will introduce an in vivo model (called the within-host model) later and the two models share several compartments and parameters that represent the same biological quantities or processes but differ in environment (i.e., ex vivo vs. in vivo), for those shared compartments and parameters, we use letters with a prime to indicate ex vivo quantities. For example, $a'_L$ is the length of ex vivo asexual life cycle and may differ from its in vivo counterpart $a_L$ due to a potential environment-induced variation, which was evidenced in *P. falciparum*[24]. Since in the ex vivo experiment the parasites taken from clinical patients may contain a small amount of schizonts that are not fully mature or ruptured, we chose the initial age distribution of the parasites in the culture

$P'(a, 0)$ to be

$$P'(a, 0) = P'_c(0) \frac{N_{(\mu', \sigma')}(a) + q N_{(\mu' + a'_L, \sigma')}(a)}{\sum \left( N_{(\mu', \sigma')}(a) + q N_{(\mu' + a'_L, \sigma')}(a) \right)} \tag{3}$$

where $N_{(\mu', \sigma')}(a)$ and $N_{(\mu' + a'_L, \sigma')}(a)$ are the probability density functions of the normal distribution $N(\mu', \sigma')$ and $N(\mu' + a'_L, \sigma')$ truncated between 0 and $a'_L$ and binned every one hour, representing the age distributions of parasites in the current and the preceding replication cycles, respectively. Since the number of parasites in the current cycle is larger than that in the preceding cycle because of parasite replication, we introduce a parameter $q \in [0, 1]$ to capture this difference (note that $q$ can be considered as the reciprocal of the parasite multiplication factor). The probability density is normalised to the sum of the probability density (denominator of Eq. 3) such that the sum of $P'(a, 0)$ over all ages is equal to the initial parasite number $P'_c(0)$ (which will be explained later by Eq. 7). The dead cells that have not ruptured in the culture follow:

$$P'_d(a, t) = P'_d(a, t - 1)e^{-r'} + P'(a, t - 1)\left(1 - e^{-\delta'}\right), a = 1, 2, 3, \ldots, a'_L, \tag{4}$$

where $r'$ is the in vitro rate of dead cell rupture.

The ex vivo experiments measured PvLDH in two compartments, PvLDH in the pellet (intracellular PvLDH) and in the media (extracellular PvLDH). The dynamics of PvLDH in the media (denoted by $L_m$) are modelled by the following difference equations:

$$
\begin{aligned}
L_m(t) = & L_m(t - 1)e^{-\lambda'} + \sum_{a=1}^{a'_L} c'(a)P'_d(a, t - 1)\left(1 - e^{-r'}\right) \\
& + c'(a'_L)P'(a'_L, t - 1)e^{-\delta'}.
\end{aligned} \tag{5}
$$

The first term on the righthand side of Eq. 5 represents a reduced amount of PvLDH compared to the amount at the preceding time $L_m(t - 1)$ due to natural degradation at rate $\lambda'$. The second term represents PvLDH released from the ruptured dead cells. The third term represents the amount of PvLDH released into plasma from the rupturing pRBCs. $c'(a)$ is the cumulative mass of PvLDH inside a single pRBC (note that the cumulative mass is a net result of PvLDH production and degradation inside the pRBC). Since it was found that high LDH activity in the regulation of the formation of pyruvate was associated with mature parasites (i.e., trophozoites and schizonts)[4], $c'(a)$ is heuristically modelled by an increasing sigmoidal function of parasite age $a$:

$$c'(a) = \frac{c_1' a^2}{a^2 + c_2'^2}, \tag{6}$$

where $c_1'$ and $c_2'$ are the tuning parameters modulating the shape of the sigmoidal curve. The initial PvLDH level in culture media $L_{m0} = L_m(0)$ is also a model parameter. The net accumulation rate of intraerythrocytic PvLDH is the derivative of $c'(a)$ with respect to $a$ and is given by $2c_1'c_2'^2 a / (a^2 + c_2'^2)^2$. Model parameters and associated information including prior distribution for the Bayesian inference (see below) are provided in Table 1. Note that the length of ex vivo life cycle is chosen to be 72 h based on that the ex vivo life cycle was estimated to be approximately 1.5 times of the in vivo life cycle which is 48 h. In detail, it was observed in the lower panels of Fig. 1 that the ex vivo duration of trophozoite stage, which is approximately the time between 5 h culture time (which is followed by a drop in the ring fraction) and 41 h culture time (which is followed by a drop in the trophozoite fraction), is 36 h. Since the duration of in vivo trophozoite stage is approximately 24 h (19 h to 42 h post-invasion in a 48 h life cycle), the ratio of ex vivo trophozoite duration to in vivo trophozoite duration is 1.5, which, if applied to all other stages (i.e., ring and schizont), results in an estimated ex vivo lifecycle of 72 h. We would like to emphasise that the development of the ex vivo model and the choice of the life cycle parameter

## Table 1 | Parameters of the ex vivo model

| Parameters of the ex vivo model | | | |
|---|---|---|---|
| Parameter | Unit | Description | Prior distribution |
| $\mu'$ | h | mean of the normal distribution defining the initial parasite age distribution | U(0, 24) |
| $\sigma'$ | h | SD of the normal distribution defining the initial parasite age distribution | U(0, 20) |
| $q$ | (unitless) | a fraction parameter | U(0, 1) |
| $c_1'$ | ng | maximum level of intraerythrocytic PvLDH | U(0, 0.01) |
| $c_2'$ | h | parasite age where the intraerythrocytic PvLDH level reaches $c_1/2$ | U(0, 72) |
| $\delta'$ | $h^{-1}$ | ex vivo death rate of parasites | U(0, 0.5) |
| $r'$ | $h^{-1}$ | ex vivo rupture rate of dead parasite | U(0, 1) |
| $\lambda'$ | $h^{-1}$ | ex vivo decay rate of PvLDH | N(0.0106, $4.1565 \times 10^{-4}$) within (0, 0.1) |
| $L_{m0}$ | ng | initial level of PvLDH in culture media | U(0, 1.755) |
| $a_L'$ | h | ex vivo length of life cycle | Fixed to be 72 |

The prior distribution for the population mean of each model parameter is a uniform distribution with biologically plausible boundaries specified in the table. Note that the ex vivo length of asexual life cycle $a_L'$ was fixed to be 72 h, as explained in the main text. The prior for $\lambda'$ was determined based on a modelling study of an independent in vitro experiment where decay data of a human PvLDH were collected and fitted by an exponential decay model (Eq. 1; see the first section of Methods for details). We assigned an upper limit of $L_{m0}$ of 1.755 ng because the initial PvLDH level in culture media was below the limit of quantification of 3.9 ng/mL, which is equivalent to 1.755 ng based on the culture media volume of 0.45 mL. Priors for other parameters were set to be uniform distributions with bounds selected either based on biological plausibility (e.g., 0 as a lower bound) or to be some appropriate values to avoid any resolvable truncation of the posterior distribution (e.g., those upper bounds shown in the table).

were done without the intention to redefine or estimate the length of the *P. vivax* life cycle for the ex vivo system, as the available data is insufficient to determine the parameter. The challenges of the *P. vivax* culture system likely contributed to the extended life cycle that we observed, including loss of viable parasites and the increase in frequency of morphologically unhealthy parasites over time. Future advancements in *P. vivax* culture will enable the use ex vivo systems that more accurately reflect the in vivo parasite life cycle.

The ex vivo model can generate the following experimentally measured quantities that are fitted to the ex vivo data:

- The number of parasites in the culture (denoted by $P_c'$) is given by the sum of $P'(a, t)$ over all ages:

$$P_c'(t) = \sum_{a=1}^{a_L'} P'(a, t), \tag{7}$$

- The PvLDH in the pellet (denoted by $L_{pe}$) is given by

$$L_{pe}(t) = \sum_{a=1}^{a_L'} c'(a)[P'(a, t) + P_d'(a, t)], \tag{8}$$

- Total PvLDH in the media is given by $L_m(t)$, which is obtained by solving Eq. 5.

    To predict the fractions of parasites in different stages (i.e., the predicted fractions of rings, trophozoites and schizonts in Fig. 1), we derive the mathematical expressions of those fractions using the ex vivo model:

- The fraction of rings (defined to be fraction *of the parasites in the* age range of 1–27 h based on the prolonged ex vivo life cycle of estimated 72 h):

$$f_r'(t) = \frac{\sum_{a=1}^{27} P'(a, t)}{\sum_{a=1}^{72} P'(a, t)}. \tag{9}$$

- The fraction of trophozoites (defined to be fraction of the parasites in the age range of 28–63 h based on the prolonged ex vivo life cycle of

estimated 72 h):

$$f_{tr}'(t) = \frac{\sum_{a=28}^{63} P'(a, t)}{\sum_{a=1}^{72} P'(a, t)}. \tag{10}$$

- The fraction of schizonts (defined to be fraction of the parasites in the age range of 64–72 h based on the prolonged ex vivo life cycle of estimated 72 h):

$$f_s'(t) = \frac{\sum_{a=64}^{72} P'(a, t)}{\sum_{a=1}^{72} P'(a, t)}. \tag{11}$$

## Volunteer infection study (VIS)

The characterisation of in vivo PvLDH dynamics was based on longitudinal parasitemia and PvLDH concentrations in eight malaria-naïve healthy male and female (non-pregnant, non-lactating) subjects who participated in a Phase 1b *P. vivax* induced blood-stage malaria clinical trial[25] conducted by the clinical unit Q-Pharm Pty Ltd at QIMR Berghofer Medical Research Institute between 2016 and 2017. The study was approved by the QIMR Berghofer Medical Research Institute Human Research Ethics Committee and the Australian Defense Human Research Ethics Committee. Study design, procedures and main results have been published elsewhere[25]. Briefly, subjects were inoculated intravenously on day 0 with approximately 564 viable *P. vivax*-infected RBCs originally isolated from a patient with *P. vivax* malaria acquired in India. Parasitemia was measured by 18S qPCR[26] and monitored with twice daily blood sampling from day 4 until treatment on days 8–10, and then more frequently until 20 days after treatment. Subjects were treated with a standard course of oral chloroquine (CQ) over 3 days on day 9 ($n = 1$) or day 10 ($n = 7$), as described in[25].

## Quansys 5-plex ELISA

PvLDH levels in VIS were measured in whole blood samples collected between day 4 and day of treatment, and every 12 hours after treatment until 120 hours. PvLDH levels in the samples above were measured using the Q-

Plex™ Human Malaria assay (Quansys Biosciences, Logan, UT, USA) as previously described[27]. The Q-View™ Imager Pro was used to capture chemiluminescent images of each plate which was then quantitatively analysed using Q-View™ software (Quansys Biosciences).

### Within-host model of *P. vivax* and PvLDH dynamics for VIS data

We developed a within-host model to capture in vivo *P. vivax* and PvLDH dynamics for the VIS. We adopted the difference equation model structure previously developed[19,28–30] to model parasite replication dynamics. In detail, let $P(a, t)$ represent the number of pRBC in the peripheral blood and depends on both parasite age $a$ and time $t$ (both in unit of hours), we have:

$$P(a, t) = \begin{cases} P(a-1, t-1)e^{-\delta-\overline{k_d}}, & a = 2, 3, \ldots, a_L, \\ \rho P(a_L, t-1)e^{-\delta-\overline{k_d}}, & a = 1 \end{cases} \quad (12)$$

where $a_L$ is the in vivo length of *P. vivax* asexual replication cycle and $t$ takes integer hours (i.e., $t = 0, 1, 2, 3, \ldots$). $\delta$ is the in vivo death rate of pRBCs. $\rho$ is the schizont-to-ring expansion factor indicating the average number of new rings formed due to the rupture of a single schizont from the previous replication cycle. Note that $\rho$ is different from the parasite multiplication factor (PMF) or parasite multiplication rate (PMR), which indicates the average amplification rate of parasite number over one asexual life cycle[31], and the relationship between $\rho$ and PMF follows PMF $= \rho e^{-\delta a_L}$. Note that the PMF will be less than the schizont-to-ring expansion factor unless the parasite death rate $\delta$ is zero. All survival parasites with age $a_L$ will rupture at the next time step. $\overline{k_b}$ is the average rate of drug-induced parasite killing during each time step (i.e., one hour) and is given by the average of instantaneous killing rate $k_d(t)$ at the boundaries of the time step from $t-1$ to $t$, i.e., $\overline{k_d} = (k_d(t-1) + k_d(t))/2$ where

$$k_d(t) = \frac{k_{\max}C(t)^{\gamma}}{C(t)^{\gamma} + EC_{50}^{\gamma}}, \quad (13)$$

$k_{max}$ is the maximum rate of parasite killing by CQ and $EC_{50}$ is the half-maximal effective concentration at which the killing rate reaches 50% of the maximum killing rate. $\gamma$ is the Hill coefficient determining the curvature of the dose-response curve. $C(t)$ represents the effective CQ concentration in the central compartment at time $t$ and a CQ concentration versus time profile for each volunteer was simulated using the pharmacokinetic model of CQ developed by Abd Rahman et al.[32]. Parameter values used in the

simulation are provided in Table 2 (the plasma samples column) of Abd Rahman et al.'s article.

The dynamics of dead pRBCs at age $a$ and time $t$, $P_d(a, t)$, are modelled by:

$$P_d(a, t) = P_d(a, t-1)e^{-r} + P(a, t-1)\left(1 - e^{-\delta-\overline{k_d}}\right), a = 1, 2, 3, \ldots, a_L, \quad (14)$$

where $r$ is the rate of dead pRBC clearance from the peripheral blood.

We assume the age distribution of the 564 pRBCs at $t = 0$ follows a normal distribution $N(\mu, \sigma)$ truncated between 0 and $a_L$ and binned by one hour, and parasites with age $a$ are those in the bin $(a - 1, a)$. $\mu$ and $\sigma$ are the mean and standard deviation of the normal distribution respectively.

For the PvLDH turnover dynamics, the mass of PvLDH (in ng) in the plasma, denoted by $L_p(t)$, increases due to the rupture of fully mature pRBCs (schizonts at the end of asexual replication cycle) to the peripheral blood and decreases due to natural degradation. The discrete model for $L_p(t)$ is given by:

$$L_p(t) = L_p(t-1)e^{-\lambda} + \left(\frac{V_p}{V_e}\right)c(a_L)P(a_L, t-1)e^{-\delta-\overline{k_d}}. \quad (15)$$

The PvLDH concentration in the plasma at $t = 0$, $L(0)$, was set to be zero (i.e., no plasma PvLDH before the inoculation with parasites). The first term on the righthand side of Eq. 15 represents a reduced amount of PvLDH compared to the amount at the preceding time $L(t-1)$ due to natural degradation at rate $\lambda$. The second term represents the amount of PvLDH released from the ruptured pRBCs to the plasma. $V_p$ and $V_e$ represent the volume of plasma and the volume of extracellular fluid (both in mL) and are given by:

$$V_p = 80W_b(1 - H), \quad (16)$$

and

$$V_e = 200W_b, \quad (17)$$

where $W_b$ is body weight and $H$ is haematocrit[10,12]. Note that we assume dead cells do not contribute to the plasma in the peripheral blood because they are quickly phagocytosed by macrophages and the enclosed PvLDH is

### Table 2 | Parameters of the within-host model

| Parameters of the within-host model | | | |
|---|---|---|---|
| **Parameter** | **Unit** | **Description** | **Prior distribution** |
| $\mu$ | h | mean of the normal distribution defining the initial parasite age distribution | U(0, 48) |
| $\sigma$ | h | SD of the normal distribution defining the initial parasite age distribution | U(0, 80) |
| $\rho$ | (unitless) | schizont-to-ring expansion factor | U(0, 100) |
| $c_1$ | ng | maximum level of intraerythrocytic PvLDH | logN(-7.15, 0.65) within (0, 0.1) |
| $c_2$ | h | parasite age where the intraerythrocytic PvLDH level reaches $c_1/2$ | N(29.17, 8.40) within (0, 72) |
| $\delta$ | h$^{-1}$ | in vivo death rate of parasites | U(0, 1) |
| $r$ | h$^{-1}$ | clearance rate of dead pRBCs from circulation | U(0, 2) |
| $\lambda$ | h$^{-1}$ | in vivo decay rate of PvLDH | U(0, 0.1) |
| $k_{max}$ | h$^{-1}$ | maximum parasite killing rate by CQ | N(0.213, 8.7 × 10$^{-3}$) within (0, 1) |
| $EC_{50}$ | ng/mL | CQ's half-maximal effective concentration | N(15, 2) within (0, 50) |
| $\gamma$ | (unitless) | Hill coefficient of the dose-response curve | Fixed to be 2.5 |
| $a_L$ | H | in vivo length of asexual life cycle | Fixed to be 48 |

The prior distributions are chosen for the population means of the model parameters. The lower bounds of the model parameters for those with a prior uniform distribution were selected based on biological plausibility (e.g., 0 as a lower bound). The priors for $c_1$ and $c_2$ are chosen to be the posterior distributions obtained from the ex vivo model fitting (see the green curves in Supplementary Fig. 1 in the Supplementary Information). The priors for $k_{max}$ and $EC_{50}$ were chosen based on the estimates from Abd Rahman et al.[32].

assumed to remain inside the macrophages. Similar to Eq. 6, the cumulative intraerythrocytic PvLDH level $c(a)$ for in vivo parasites is given by:

$$c(a) = \frac{c_1 a^2}{a^2 + c_2^2},\tag{18}$$

where $c_1$ and $c_2$ are the tuning parameters modulating the shape of the sigmoidal curve. The net accumulation rate of intraerythrocytic PvLDH is the derivative of $c(a)$ with respect to $a$ and is given by $2c_1c_2^2 a/(a^2 + c_2^2)^2$.

The mass of whole blood PvLDH (denoted by $L_h(t)$) is given by the sum of plasma PvLDH $L_p(t)$ and the mass of PvLDH inside either viable parasites or dead pRBCs:

$$L_h(t) = L_p(t) + \sum_{a=1}^{a_L} c(a)P(a,t) + \sum_{a=1}^{a_L} c(a)P_d(a,t).\tag{19}$$

Model parameters and associated information including prior distributions for Bayesian inference are provided in Table 2.

The within-host model can generate several experimentally measured quantities that will be fitted to the VIS data. The number of circulating pRBCs per mL of peripheral blood (i.e., the peripheral blood parasitemia, denoted by $P_c(t)$) is given by:

$$P_c(t) = \sum_{a=1}^{a_L} \frac{P(a,t)}{V_b},\tag{20}$$

where $V_b$ is the whole blood volume (mL) and is estimated based on 80 mL per kg of the volunteer's body weight, i.e., $V_b = 80W_b$[10]. Note that the initial parasitemia $P_{c0} = P_c(0)$ is fixed to be 564 based on the experimental inoculation size. $L_h(t)$ was converted from mass to concentration by dividing it by the total blood volume $V_b$, in order to fit to the whole blood PvLDH measurements which are given by concentration.

To predict the fractions of parasites in different stages (i.e., the predicted fractions of rings, trophozoites and schizonts in Fig. 3), similar to what we did with the ex vivo model, we derive the mathematical expressions of those fractions using the within-host model. The fraction of rings (defined to be fraction of the parasites in the age range of 1–18 h based on the in vivo life cycle of 48 h):

$$f_r(t) = \frac{\sum_{a=1}^{18} P(a,t)}{\sum_{a=1}^{48} P(a,t)}.\tag{21}$$

The fraction of trophozoites (defined to be fraction of the parasites in the age range of 19–42 h based on the in vivo life cycle of 48 h)

$$f_{tr}(t) = \frac{\sum_{a=19}^{42} P(a,t)}{\sum_{a=1}^{48} P(a,t)}.\tag{22}$$

The fraction of schizonts (defined to be fraction of the parasites in the age range of 43–48 h based on the in vivo life cycle of 48 h)

$$f_s(t) = \frac{\sum_{a=43}^{48} P(a,t)}{\sum_{a=1}^{48} P(a,t)}.\tag{23}$$

### Relationship between inhouse ELISA PvLDH data and Quansys PvLDH data

The posterior distributions of $c_1'$ and $c_2'$ obtained from the ex vivo model fitting was used as the prior distributions of $c_1$ and $c_2$ in the within-host model. To perform this, we established the relationship between PvLDH concentrations generated using inhouse ELISA and Quansys methods, given that PvLDH concentrations were measured by inhouse ELISA in the ex vivo system while concentrations in the VIS were measured by Quansys. Using this relationship, we transformed the VIS PvLDH data to calibrate with the ex vivo data measured by inhouse ELISA. The relationship between

inhouse ELISA and Quansys PvLDH concentrations was established using a retrospective cross-sectional dataset of 58 vivax malaria patients in Sabah, Malaysia, who had whole blood PvLDH concentrations measured by both inhouse ELISA and Quansys methods[27]. Using a Bayesian approach, we fitted a linear model to the log-transformed ELISA and Quansys data from Malaysian patients (Supplementary Fig. 6 in the Supplementary Information). The relationship was estimated to be:

$$\log(\text{ELISA PvLDH}) = 0.96 \times \log\big(\text{multiplexed ELISA PvLDH}\big) + 2.04,\tag{24}$$

which was used to calibrate the VIS PvLDH concentrations to the ex vivo ELISA measurements. The Malaysian patient data and computer code for the model fitting are publicly available at https://doi.org/10.26188/23256413.v1.

### Statistics and reproducibility
Bayesian hierarchical modelling was performed for model fitting and parameter estimation for the ex vivo model and the within-host model, because the experimental data are grouped by patients. The hierarchical model expands the mathematical model (either the ex vivo model or the within-host model) by introducing two levels of model parameters. The individual-level parameters (or simply the individual parameters) are the model parameters applicable to individuals such as patient isolates or volunteers. Each individual owns a set of model parameters that differs from that of other individuals. For example, each of the eight volunteers in VIS has 10 parameters (i.e., those described in Table 2) to determine their own infection kinetics and the 10 parameters for one volunteer differ from the 10 parameters for another volunteer. Therefore, there are in total 80 individual parameters in the hierarchical within-host model. The population-level parameters (or simply the population parameters) determine how the individual parameters are distributed. We assume individual parameters follow a multivariate normal distribution with a set of means (referred to as the population mean parameters) and standard deviations (referred to as the population SD parameters). Note that since the parameters in our models are positive and bounded due to biological plausibility, we assume that log-transformed individual parameters follow a multivariate normal distribution with log-transformed population means and SDs.

With a hierarchical model (either the ex vivo model or the within-host model), fitting the model to experimental data (either the ex vivo data or the VIS data) and sampling from the posterior distribution for parameter estimation were implemented in R (version 4.0.5) and Stan (RStan 2.21.5) using the Hamilton Monte Carlo sampling method optimized by the No-U-Turn Sampler[33,34]. Four chains were randomly initiated, and each chain generated 1000 (for the ex vivo model fitting) or 2000 (for the within-host model fitting) samples (excluding burn-in samples), giving in total 4000 (for the ex vivo model fitting) or 8000 (for the within-host model fitting) samples drawn from the posterior distribution. Note that the longer chains for within-host model fitting were required to increase effective sample size. The M3 method was used to penalise the likelihood for data below the limit of detection[35]. Diagnostic outputs, such as $\hat{R}$ statistic and effective sample size were examined to ensure the convergence of the chains and low sample autocorrelations. While it is not necessary/beneficial to perform hierarchical modelling for the existing ex vivo data where only two patients were involved, the method can be directly applied to new data with a larger sample size when the data is available.

The median model prediction and 95% prediction interval (PI) are given by the median and quantiles of 2.5% and 97.5% of the 4000/8000 model solutions at each time respectively. For example, to produce the model prediction shown in Fig. 3a, we simulated the within-host model 8000 times using the 8000 posterior values of the individual parameters of the within-host model for Volunteer R009 and calculated the median and 95% PI for all time points. The median and 95% credible interval (CrI) of a model parameter (e.g., the PvLDH decay rate) are given by the median and quantiles of 2.5% and 97.5% of the posterior samples of the parameter. To

estimate the in vivo (or ex vivo) intraerythrocytic PvLDH level, posterior samples of the population mean of $c_1$ and $c_2$ (or $c_1'$ and $c_2'$ for the ex vivo model) were put into Eq. 18 (or Eq. 6 for the ex vivo model) to obtain the median and 95% PI for all parasite age values (see Fig. 2 for results). Similarly, the ex vivo/in vivo net PvLDH accumulation rate can be calculated using the same posterior samples of the population mean parameters and the derivatives of Eqs. 6 and 18.

All the fitting and simulation results are reproducible by using the data and code publicly available at https://doi.org/10.26188/23256413.v1.

## Data availability

All main text figures (Fig. 1–4) and Supplementary Figs. were generated using MATLAB (version 2019b; The MathWorks, Natick, MA). The source data underlying Figs. 1 and 2 can be found in the files "Exvivo_data.xlsx" and "VIS_data.xlsx" at https://doi.org/10.26188/23256413.v1.

## Code availability

All experimental data, computer code (R and MATLAB) for model fitting and figure generation are publicly available at https://doi.org/10.26188/23256413.v1.

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

## Acknowledgements

We thank Dr Sophie Zaloumis (University of Melbourne) for helpful conversations about the method for model fitting and Dr Louise Marquart (University of Queensland) for her preliminary analysis of the VIS data. We also thank Dr Gonzalo Domingo (PATH) for his contribution to data collection and analysis. The work was supported by the National Health and Medical Research Centre (NHMRC) of Australia: Project Grant (1025319), a Senior Principal Research Fellowship to Nicholas M. Anstey (1135820), Investigator Grants Leadership Level to James McCarthy (GNT2016396) and Julie A Simpson (1196068) and supported in part by the Australian Centre for Research Excellence on Malaria Elimination (1134989). Steven Kho is supported by a Menzies Future Leaders Fellowship.

## Author contributions

N.M.A., J.S.M., S.B. conceived the study; S.K., M.J.G., B.E.B., K.A.P., T.W., J.R.P., I.K.J. conducted the experiments and provided the data; P.C., S.K., S.B. analysed the data with contributions from J.A.S., J.M.M., N.M.A., J.S.M.; P.C. developed the mathematical models and performed model fitting, simulation and analysis; P.C. and S.K. wrote the first draft of the manuscript with contributions from all other authors. All authors reviewed and commented on the manuscript.

## Competing interests

The authors declare no competing interests.
