## [Peer Review File · Communications Biology]

Reviewers' comments:

Reviewer #1 (Remarks to the Author):

This study combines data from Controlled Human Malaria Infection studies with mathematical models to investigate the kinetics of Plasmodium vivax lactate dehydrogenase (LDH) – a key biomarker for rapid tests for P. vivax. The CHMI experiments appear to have been implemented in a thorough and robust manner by an experience team of investigators, and the mathematical models are sensibly developed with appropriate statistical inference procedures (with one possible exception that I'll outline below). Overall, I really liked this manuscript, and I think it makes a useful contribution to the literature – most notably in the provision of an estimate of the half-life of LDH antigen. My comments are minor and are primarily related to the logical structure of the manuscript.

Model building

The manuscript presents data and models from in vitro, ex vivo, and in vivo experiments. Upon first reading, it's not clear how these components are linked. For example, the in vitro experiment and model is only mentioned very late in the Materials and Methods section. Upon further reading it becomes evident that the models for these data sets are linked in terms of their Bayesian priors and posteriors. For those less familiar with Bayesian model building, it would be good to provide an overview of how the experiments are linked.

Apart from their role in providing prior information to the in vivo model, I believe that the parameters estimated from the ex vivo and in vitro models are of limited value. These are likely to be highly dependent on the peculiarities of the experimental set up, and may not transfer to other laboratories.

Statistical inference

I was very impressed with the hierarchical Bayesian statistical inference procedure for the in vivo model – I would consider this to be state of the art stuff. What was less clear was the statistical inference procedure utilised for the ex vivo model – fits as shown in Figure 1. Was this also a hierarchical model? – difficult with only two participants. Were models fitted separately to the two participants? Or were all parameters shared? In particular, the model fits to the ring/troph/schizont data appear to miss some important qualitative behaviour.

In vivo study participants

These participants were recruited from a malaria vaccine trial. Could the authors please specify if participants were from the control or vaccine arms, and what type of vaccine this was? For example, if this was a blood-stage P. vivax vaccine, there could be very important effects.

Descriptive statistics

This paper uses a sophisticated model building procedure to provide an estimate of the PvLDH half-life. What estimate would be obtained if the authors took the much simpler approach of just fitting a linear regression to the post-treatment PvLDH data? This question is primarily for my own curiosity, and can be ignored for the revision of the manuscript.

Reviewer #2 (Remarks to the Author):

This important study characterises both ex vivo and in vivo PvLDH dynamics, providing important data on P. vivax biology and the development of PvLDH-based diagnostic tools (which are currently the major form of POC rapid diagnostics for vivax malaria).

The data from the in vitro and ex vivo studies are individually important and valid data sets; however,

several confounding factors necessitate caution when drawing conclusions from the comparison of these data sets. These confounders are as follows (some identified by the authors in the discussion, some important ones are not):

1. Firstly (as mentioned by the authors) the genetic background and phenotype of Indian and Papuan *P. vivax* should be considered. *P. vivax* populations are highly diverse and the Indian and Papuan populations are likely to have been isolated by 100s of thousands of years. Phenotypes such as PLDH expression (see below 1a), microvesicular formation, time of asexual erythrocytic development, vir expression (cytoadhesion) and rosetting rates (see below 1b.) have not been compared side by side in an ex vivo manner. Below I expand on two of these points:

1a. PLDH: The intrinsic production of PLDH by Indian and Papuan isolates should have been compared side by side, failing that, perhaps sequencing the PLDH and copy # analysis for the Indian Pv versus the Papuan Isolates. One point that minimises this criticism, is that a 2020 study by Lee et al (Malar J. 2020 Feb 4;19(1):60. doi: 10.1186/s12936-020-3134-y. PMID: 32019541; PMCID: PMC7001217.) showed that PLDH is relatively conserved.

1b. Rosetting: While rosetting rates (an important feature of Pv biology more prevalent in Pv than Pf) are unlikely to affect the PLDH expression ex vivo, it is likely to profoundly affect the modelling of parasite numbers/stage in vivo, as rosetting Pv is more likely to sequester in the spleen and microvasculature, skewing the peripheral assessment of parasitaemia (we really can't tell what proportion of stages will be present in vivo). Has anyone investigated rosetting rates in the Indian isolates?

2. The comparison of antigen kinetics in an in vivo and ex vivo, is fraught with a number of confounding factors affecting parasite/PLDH expression, including but not limited to: i. the composition of the growth media (20% Serum ex vivo, 50% in vivo... gentamycin (which we know to cause a delayed death phenotype in Pv), glucose levels ect); ii. The presence of in vivo immune factors (WBCs, Platelets and Complement ect) that affect the integrity of the IRBC membrane and rate of rupture/exocytosis; iii. The in vivo system is open system vs the closed ex vivo system where nutrients/toxins (affecting parasite health) are not cleared/turned over in a continuous manner. Usually ex vivo studies (antimalarial sensitivity assays, invasion assays ect) are internally controlled to account for these factors. Of course, some of these factors (mentioned in the discussion) are the very reason for why the authors conclude it is not possible to use the in vivo method to examine PLDH dynamics ("these results highlight the necessity of using in vivo estimates of PvLDH dynamics in any future predictive modelling") which in many ways is circular reasoning. Perhaps what is needed is a more carefully controlled comparison (conducting the ex vivo work using the samples collected from the Australian in vivo studies).

3. Host issues: In vivo work was carried out using healthy, non-immune, adult volunteers with no underlying RBC mutations or anaemia; compared to ex vivo blood/IRBCs from hosts with almost certainly a different genetic background (from a region with a high rate of RBC mutations). These factors may be very important when making comparisons.

While the above confounding factors do not impact on the individual validity of each set of data (in vivo and ex vivo), it does make it difficult to make value claims (ie. One model is of more value than the other; and if so, under all circumstances?). In fact the recent work on rosetting and Pv sequestration in organs such as the spleen make it difficult to accurately model parasitaemias from peripheral samples and relate these to PLDH levels collected in geographically/genetically distinct populations (esp. as the architecture of spleens and immune response differs greatly between volunteers from say a holo/hyper endemic region vs Healthy Aussies). While I do not suggest that the authors conduct any additional experiments, I do hope they consider moderating their conclusions. On another point; I am a little confused at the modelled 72h life cycle of ex vivo isolates? Are the authors suggesting that the intra-erythrocytic life cycle of ex vivo *P. vivax* (invasion of merozoite into retic and development to segmenting/bursting schizont is 72hrs? if so, this would be in stark contrast to the now extensive body of literature on *P. vivax* ex vivo biology? and if not, please better explain what is meant by a 72hr life cycle?)

Overall, an excellent body of work by leaders in the field of in vivo and ex vivo biology, certainly their respective data sets provide important information for a wide range of scientists/clinicians/diagnostic

developers. I suggest the authors modify some of the conclusions and provide suggestions for future studies. I am very glad to answer any queries regarding my review and I do apologise in advance if I have misunderstood any of the points made by the authors.

Reviewers' comments:

Reviewer #1 (Remarks to the Author):

This study combines data from Controlled Human Malaria Infection studies with mathematical models to investigate the kinetics of Plasmodium vivax lactate dehydrogenase (LDH) – a key biomarker for rapid tests for P. vivax. The CHMI experiments appear to have been implemented in a thorough and robust manner by an experience team of investigators, and the mathematical models are sensibly developed with appropriate statistical inference procedures (with one possible exception that I'll outline below). Overall, I really liked this manuscript, and I think it makes a useful contribution to the literature – most notably in the provision of an estimate of the half-life of LDH antigen. My comments are minor and are primarily related to the logical structure of the manuscript.

Model building

The manuscript presents data and models from in vitro, ex vivo, and in vivo experiments. Upon first reading, it's not clear how these components are linked. For example, the in vitro experiment and model is only mentioned very late in the Materials and Methods section. Upon further reading it becomes evident that the models for these data sets are linked in terms of their Bayesian priors and posteriors. For those less familiar with Bayesian model building, it would be good to provide an overview of how the experiments are linked.

Thank you for the suggestion. We have revised the manuscript to explain how different pieces of modelling work are related in the revised manuscript:

- Line 79—90 (in Introduction)
“In this study, we quantified the dynamics of PvLDH by fitting two mathematical models to experimental data. The first model (referred to as the *ex vivo* model) captures the *ex vivo* dynamics of parasites and PvLDH and was fitted to a set of longitudinal measurements of parasite count and PvLDH concentration in two short-term *ex vivo* cultures of *P. vivax* patient isolates. The parameters identified from the *ex vivo* model then served as prior knowledge for the second model (referred to as the within-host model), which captured *in vivo* parasite and PvLDH dynamics that was fitted to longitudinal data of parasitemia and whole blood PvLDH concentration from eight adults experimentally-infected with *P. vivax* in a volunteer infection study (VIS). Both model fittings were conducted using Bayesian hierarchical inference, which is a statistical method for generating the posterior distribution of model parameters that quantify the mean properties of the examined cohort, based on both experimental data and any prior knowledge of the parameter distribution gained from other independent experimental studies.”
- Line 177—181 (in Results)
“To refine the within-host model, priors of two model parameters that determine the intraerythrocytic PvLDH level (i.e., c_1 and c_2 in the within-host model; see Table 2 in the Materials and Methods) were proposed based on the posterior distributions of the *ex vivo* model (see the green dotted curves in Fig. S1 in the Supplementary Information).”

To clarify this further, we have added a short section at start of the Materials and Methods on the study design (Line 322—329 in the revised manuscript):

“The mathematical model of *P. vivax* and PvLDH dynamics was developed in a two-step process. First, we developed the *ex vivo* model which was fitted to the *ex vivo* experimental data derived from *P. vivax* culture and serial measurements of PvLDH at different lifecycle stages during the *ex vivo* culture. Parameters identified by this model were then combined with longitudinal parasitemia and whole blood PvLDH measurements from a *P. vivax* malaria volunteer infection study (VIS) to develop the within-host (*in vivo*) model in order to calculate key biomarker dynamics.”

Apart from their role in providing prior information to the *in vivo* model, I believe that the parameters estimated from the *ex vivo* and *in vitro* models are of limited value. These are likely to be highly dependent on the peculiarities of the experimental set up, and may not transfer to other laboratories.

We agree with the reviewer that the *ex vivo* and *in vitro* models are tailored to our experimental set up and conditions. This has been added as a limitation in the Discussion (line 308—311):

“Lastly, the estimates of PvLDH dynamics generated based on our experimental set up may vary between *P. vivax* geographical locations and experimental conditions. Promising advancements to enable continuous *P. vivax* culture systems are being made and will enable standardisation of future *ex vivo* studies of this Plasmodium species.”

We have also enriched the Discussion describing the likely minimal variation in parameter estimates that is potentially driven by isolates from different geographical locations (lines 296—301):

“We were unable to test whether parasite genetic factors also play a role in varying PvLDH production rates (i.e., Indian VIS isolate vs. Indonesian clinical isolates). However, recent work suggests PvLDH is genetically well-conserved in global populations¹⁸, and that the pattern of our modelling estimates was consistent with transcriptomic data from Thai *P. vivax* isolates¹⁹, suggesting minimal genetic variation.”

Statistical inference

I was very impressed with the hierarchical Bayesian statistical inference procedure for the *in vivo* model – I would consider this to be state of the art stuff. What was less clear was the statistical inference procedure utilised for the *ex vivo* model – fits as shown in Figure 1. Was this also a hierarchical model? – difficult with only two participants. Were models fitted separately to the two participants? Or were all parameters shared? In particular, the model fits to the ring/troph/schizont data appear to miss some important qualitative behaviour.

Thank you for the very positive comments on the statistical method used for fitting the model to the data. We confirm that the fitting of the *ex vivo* model to data was also performed using Bayesian hierarchical inference. We used the same Bayesian framework for both *ex vivo* and *in vivo* model fitting to ensure methodological consistency. But we agree that hierarchical modelling may not necessarily be superior to fitting the patients separately for a very small sample size (like 2 in the *ex vivo* modelling study), and we expect that the established modelling framework will help future experimental design and ultimately improve estimation of population-level parameters.

We agree with the reviewer that some of the parasite staging behaviour in the *ex vivo* system was not precisely predicted, such as the transition from ring to trophozoite in both patient

isolates (Fig 1E). Our model was designed to capture the primary outcome of PvLDH dynamics over time, and was not refined to reproduce the staging transitions. The model predictions in Fig 1E does not affect the quality of the model fits to parasite count and PvLDH data and, thus, has little effect on the estimates of PvLDH dynamics which depend on parasite age in time rather than how the parasite age is classified into different stages. Further investigation of the staging process requires the development of new models and possibly additional data for experimental validation and thus are left for future work. We have added this as a limitation in the discussion (line 301—308 in the revised manuscript):

“Fourthly, we found that the *ex vivo* model predictions did not precisely reproduce some of the parasite staging behaviour seen in the data, such as the ring-to-trophozoite transition in both patients. This suggests that the staging process was either not captured by the current model (which was not designed for studying the staging process) or not informed by the parasitemia data. Nevertheless, this did not affect the quality of the primary data on model-predicted PvLDH dynamics in our study. Further work is warranted to investigate the staging process which may require the development of new models and additional experimental validation.”

In vivo study participants

These participants were recruited from a malaria vaccine trial. Could the authors please specify if participants were from the control or vaccine arms, and what type of vaccine this was? For example, if this was a blood-stage *P. vivax* vaccine, there could be very important effects.

The volunteer infection study (VIS) from which participants were recruited for our study was a phase 1b malaria study to establish a clinical *P. vivax* transmission model using a new *P. vivax* strain and treated with chloroquine to induce gametocytemia. This was not a malaria vaccine trial. All participants for our study were inoculated with a standardised inoculum of parasitised red cells and treated with chloroquine at days 9-10.

Descriptive statistics

This paper uses a sophisticated model building procedure to provide an estimate of the PvLDH half-life. What estimate would be obtained if the authors took the much simpler approach of just fitting a linear regression to the post-treatment PvLDH data? This question is primarily for my own curiosity, and can be ignored for the revision of the manuscript.

We agree it interesting to check this and have fitted a simple exponential (or linear after log-transform of whole blood PvLDH) decay model to the post-treatment PvLDH data (fitting results are shown in the figure below). Bayesian hierarchical modelling was used to be consistent with the work presented in the manuscript.

Based on the posterior samples of the population mean parameter, WB PvLDH decay rate is estimated to have a median of 0.0468/h (95%CI: 0.0397—0.0553/h), which is larger than the PvLDH decay rate of 0.0316/h (0.0232—0.0416/h) estimated using the within-host model. The reason for the difference is that the PvLDH decay rate defined in the within-host model is not the same as the observed decay rate (i.e., the decay rate defined in the simple exponential decay model). The former quantifies the decay of free PvLDH in the plasma (which cannot be observed but has to be inferred by the data) while the latter is a combined effect of both PvLDH release from the alive and dead parasites and plasma PvLDH decay. Since dead parasites are removed from circulation faster than the decay of WB PvLDH (see Fig. 3A—D or Fig. S2 in the Supplementary Information), the unobservable plasma PvLDH decay rate should be less than the observed PvLDH decay rate, and the above fitting result confirms this.

Our work also highlighted that plasma PvLDH data would improve the estimation of the decay parameter and we discussed the importance of developing a reliable method for measuring plasma PvLDH in the manuscript (now lines 272—278 in the revised manuscript):

“This model provides a platform to further evaluate the utility of *P. vivax* biomarkers for non-invasively quantifying the hidden parasite biomass seen with this infection^{7,8,13,14}. This would require the availability of sufficiently sensitive and accurate assays for measuring PvLDH, and other potential *P. vivax* specific biomarkers, in plasma as well as whole blood. Serial measurements of plasma and whole blood PvLDH and parasitemia in patients with symptomatic malaria infection as well as in patients with asymptomatic sub-patent infection would provide additional key data to improve the accuracy of the model.”

Reviewer #2 (Remarks to the Author):

This important study characterises both *ex vivo* and *in vivo* PvLDH dynamics, providing important data on *P. vivax* biology and the development of PvLDH-based diagnostic tools (which are currently the major form of POC rapid diagnostics for vivax malaria).

The data from the *in vitro* and *ex vivo* studies are individually important and valid data sets; however, several confounding factors necessitate caution when drawing conclusions from the comparison of these data sets. These confounders are as follows (some identified by the authors in the discussion, some important ones are not):

Thank you for the opportunity to clarify these points above. We agree with the reviewer that the comparison of PvLDH dynamics in isolates from different geographical locations needs to be interpreted with caution (ie. Figure 5). The primary purpose of the *ex vivo* model was to advance our quantitative understanding of *ex vivo* PvLDH dynamics and to provide prior knowledge, despite its limitations, to facilitate the estimation of within-host model parameters. Thus, our intention was to provide complementary data, and not to compare *ex vivo* and *in vivo* PvLDH dynamics generated from two different settings as perceived in our original manuscript by both reviewers. We have now revised to exclude such comparisons, including removal of Figure 5, as well as adjustments to the text to expand on potential confounding factors affecting *ex vivo* and *in vivo* estimates. These are detailed further in the responses below.

1. Firstly (as mentioned by the authors) the genetic background and phenotype of Indian and Papuan *P. vivax* should be considered. *P. vivax* populations are highly diverse and the Indian and Papuan populations are likely to have been isolated by 100s of thousands of years. Phenotypes such as PLDH expression (see below 1a), microvesicular formation, time of asexual erythrocytic development, *vir* expression (cytoadhesion) and rosetting rates (see below 1b.) have not been compared side by side in an *ex vivo* manner. Below I expand on two of these points:

1a. PLDH: The intrinsic production of PLDH by Indian and Papuan isolates should have been compared side by side, failing that, perhaps sequencing the PLDH and copy # analysis for the Indian Pv versus the Papuan Isolates. One point that minimises this criticism, is that a 2020 study by Lee et al (Malar J. 2020 Feb 4;19(1):60. doi: 10.1186/s12936-020-3134-y. PMID: 32019541; PMCID: PMC7001217.) showed that PLDH is relatively conserved.

We agree with the reviewer on the limitations of testing *P. vivax* isolates from two geographically distinct sources, and that it would have been ideal to test the Indian and Papuan *P. vivax* isolates side by side. Our access to only fresh Papuan isolates for short-term culture and our existing complex regulatory approval of the VIS in Brisbane using Indian isolates meant that these limitations were inevitable within our network. Unfortunately, we are unable to conduct additional testing including sequencing, transcriptomics or cell culture on both isolate sources, and it was not possible to cryopreserve the Papuan isolates used for culture in this study. Nevertheless, we agree with the reviewer that while *P. vivax* isolates are highly diverse, PvLDH is relatively conserved. We have expanded on this limitation in the Discussion (lines 296—301):

“We were unable to test whether parasite genetic factors also play a role in varying PvLDH production rates (i.e., Indian VIS isolate vs. Indonesian clinical isolates). However, recent work suggests PvLDH is genetically well-conserved in global populations¹⁸, and that the pattern of our modelling estimates was consistent with transcriptomic data from Thai *P. vivax* isolates¹⁹, suggesting minimal genetic variation.”

1b. Rosetting: While rosetting rates (an important feature of Pv biology more prevalent in Pv than Pf) are unlikely to affect the PLDH expression *ex vivo*, it is likely to profoundly affect the modelling of parasite numbers/stage *in vivo*, as rosetting Pv is more likely to sequester in the spleen and microvasculature, skewing the peripheral assessment of parasitaemia (we really can't tell what proportion of stages will be present *in vivo*). Has anyone investigated rosetting rates in the Indian isolates?

We agree with the reviewer that rosetting may be a potential mechanism of *P. vivax* biomechanical retention in the spleen from their increased rigidity. To our knowledge rosetting has not been examined in Indian isolates. The main point raised by the reviewer that *P. vivax* staging is skewed in peripheral blood is noted. However, we were unable to model this given the lack of a) staging data in the VIS, and b) data on rosetting rates in *P. vivax* Indian isolates.

We believe that the advantage of our model is that as future studies contribute to the literature on rosetting rates in Indian isolates, this information can be used to refine the model to better inform *P. vivax* parasite dynamics.

2. The comparison of antigen kinetics in an *in vivo* and *ex vivo*, is fraught with a number of confounding factors affecting parasite/PLDH expression, including but not limited to: i. the composition of the growth media (20% Serum *ex vivo*, 50% *in vivo*... gentamycin (which we know to cause a delayed death phenotype in Pv), glucose levels ect); ii. The presence of *in vivo* immune factors (WBCs, Platelets and Complement ect) that affect the integrity of the IRBC membrane and rate of rupture/exocytosis; iii. The *in vivo* system is open system vs the closed *ex vivo* system where nutrients/toxins (affecting parasite health) are not cleared/turned over in a continuous manner. Usually *ex vivo* studies (antimalarial sensitivity assays, invasion assays ect) are internally controlled to account for these factors. Of course, some of these factors (mentioned in the discussion) are the very reason for why the authors conclude it is not possible to use the *in vivo* method to examine PLDH dynamics (“these results highlight the necessity of using *in vivo* estimates of PvLDH dynamics in any future predictive modelling”) which in many ways is circular reasoning. Perhaps what is needed is a more carefully controlled comparison (conducting the *ex vivo* work using the samples collected from the Australian *in vivo* studies).

We thank the reviewer for summarising further potential reasons for the difference in PvLDH dynamics between *ex vivo* and *in vivo* models. As stated above, the primary purpose of the *ex vivo* model was to advance our quantitative understanding of *ex vivo* PvLDH dynamics in relation to parasitemia and to provide prior knowledge, despite its limitations, to enable refinement of the *in vivo* model.

We have removed Figure 5, which primarily leads to the impression of a direct comparison, and expanded our Discussion with these details to as follows (lines 246—253):

“Growing *P. vivax* in culture remains a major challenge¹⁷ as reflected by the presence of morphologically unhealthy parasites and prolonged *ex vivo* maturation that is commonly seen in such experiments. In comparison to an open *in vivo* system where nutrient and serum levels are high with constant turnover of toxins, we hypothesise that metabolic activity of parasites in our closed culture system may be impaired, including PvLDH production, due to factors such as reduced serum levels and build-up of toxic by-products. Other biological conditions or

factors that are present/absent *in vivo* versus *ex vivo* may further modulate the production of intraerythrocytic PvLDH.”

3. Host issues: *In vivo* work was carried out using healthy, non-immune, adult volunteers with no underlying RBC mutations or anaemia; compared to *ex vivo* blood/IRBCs from hosts with almost certainly a different genetic background (from a region with a high rate of RBC mutations). These factors may be very important when making comparisons.

While the above confounding factors do not impact on the individual validity of each set of data (*in vivo* and *ex vivo*), it does make it difficult to make value claims (ie. One model is of more value than the other; and if so, under all circumstances?). In fact the recent work on rosetting and Pv sequestration in organs such as the spleen make it difficult to accurately model parasitaemias from peripheral samples and relate these to PLDH levels collected in geographically/genetically distinct populations (esp. as the architecture of spleens and immune response differs greatly between volunteers from say a holo/hyper endemic region vs Healthy Aussies). While I do not suggest that the authors conduct any additional experiments, I do hope they consider moderating their conclusions.

We thank the reviewer for highlighting these additional important limitations to our study design. We have undertaken significant modification to our discussion of these limitations, removed such claims and Figure 5 to highlight that our intention is not to compare *ex vivo* and *in vivo* estimates and have moderated our concluding paragraph to highlight the generation of two model estimates rather than their direct comparison (line 313—318 in the revised manuscript).

On another point; I am a little confused at the modelled 72h life cycle of *ex vivo* isolates? Are the authors suggesting that the intra-erythrocytic life cycle of *ex vivo* *P. vivax* (invasion of merozoite into retic and development to segmenting/bursting schizont is 72hrs? if so, this would be in stark contrast to the now extensive body of literature on *P. vivax ex vivo* biology? and if not, please better explain what is meant by a 72hr life cycle?)

We agree that the 72h life cycle used in the *ex vivo* model is longer than the well-documented *P. vivax* life cycle of about 48h described by Chotivanich et al. (Trans R Soc Trop Med Hyg, 95(6)677-680, 2001) and Kerlin et al. (PLoS Negl Trop Dis, 6(8):e1772, 2012). We used 48h life cycle for the within-host study, which was described in the Materials and Methods. The reasons for choosing a longer life cycle for the *ex vivo* model (noting that we used 48h for the within-host model) are twofold:

Firstly, the staging data suggested a prolonged trophozoite stage by approximately 50% (see Figs. 1E and 1F) compared to the approximately 24h trophozoite duration (i.e., 19-42h post-invasion) in a 48h life cycle. With no additional data to determine the life cycle, we assumed that the entire life cycle was also prolonged by the same extent, resulting in an assumed life cycle of 72h.

Secondly, we found that the model with a shorter life cycle would either fail to fit to the data (e.g., a life cycle less than 56h made the model unable to fit to the data by causing an infinite likelihood in the algorithm) or poorly predicted the staging data (e.g., model-predicted fractions of parasite stages were very inconsistent with the data for a life cycle between 56—60h; see the figures below).

The development of the *ex vivo* model and the choice of the life cycle parameter were done without the intention to redefine or estimate the length of the *P. vivax* life cycle for the *ex vivo* system, as the available data is insufficient to determine the parameter. The challenges of the

P. vivax culture system likely contributed to the extended life cycle that we observed, including loss of viable parasites and the increase in frequency of morphologically unhealthy parasites over time. Future advancements in *P. vivax* culture will enable the use *ex vivo* systems that more accurately reflect the *in vivo* parasite life cycle.

We have included the following discussion in the revised manuscript (lines 470—477) to provide clarity on the length of the *ex vivo* life cycle that was used:

“We would like to emphasise that the development of the *ex vivo* model and the choice of the life cycle parameter were done without the intention to redefine or estimate the length of the *P. vivax* life cycle for the *ex vivo* system, as the available data is insufficient to determine the parameter. The challenges of the *P. vivax* culture system likely contributed to the extended life cycle that we observed, including loss of viable parasites and the increase in frequency of morphologically unhealthy parasites over time. Future advancements in *P. vivax* culture will enable the use *ex vivo* systems that more accurately reflect the *in vivo* parasite life cycle.”

Fitting results using a life cycle of 56h. The fitting process is the same as that used to generate Fig. 1 in the main text except a different life cycle length and different lengths of ring/troph/schizont stages for model prediction (we chose ring 0-21h, troph 22-49h, schizont 50-56h).

Fitting results using a life cycle of 60h. The fitting process is the same as that used to generate Fig. 1 in the main text except a different life cycle length and different lengths of ring/troph/schizont stages for model prediction (we chose ring 0-22h, troph 23-52h, schizont 53-60h).

Overall, an excellent body of work by leaders in the field of in vivo and ex vivo biology, certainly their respective data sets provide important information for a wide range of scientists/clinicians/diagnostic developers. I suggest the authors modify some of the conclusions and provide suggestions for future studies. I am very glad to answer any queries regarding my review and I do apologise in advance if I have misunderstood any of the points made by the authors.

We thank the reviewers for all their valuable comments that have significantly improve the manuscript. We hope that the revised manuscript has addressed each comment and is now suitable for publication.

REVIEWERS' COMMENTS:

Reviewer #1 (Remarks to the Author):

I'm happy with the authors' responses to my previous comments.

I still have a mild difference of opinion on the appropriateness of using a hierarchical model with $n=2$. I would personally just have fitted two separate models. Please note that I'm not making any suggestion that the authors make any changes or even acknowledge this point.

Congrats on a nice study.

Reviewer #2 (Remarks to the Author):

An excellent study, thank you for considering my minor comments
Congratulations